# Factors influencing gestational diabetes self-care among pregnant women in a Syrian Refugee Camp in Jordan

**Enas A. Assaf**[1]*, **Haleama Al Sabbah**[2], **Aaliyah Momani**[1], **Rasmieh Al-Amer**[3], **Ghada A. Al-Sa'ad**[4], **Anas Ababneh**[3]

**1** Faculty of Nursing, Applied Science Private University, Amman, Jordan, **2** College of Health Sciences, Abu Dhabi University, Khalifa City, United Arab Emirates, **3** Faculty of Nursing, Yarmouk University, Irbid, Jordan, **4** United Nation High Commissioner for Refugees (UNHCR) Jordan Office, Amman, Jordan

\* e_assaf@asu.edu.jo

**Data Availability Statement:** All relevant data are within the manuscript and its Supporting Information files.

## Abstract

### Aim

The main objective of this study is to identify the level of self-care practices and the determinants of Gestational Diabetes Mellitus (GDM) among pregnant women residing in one of the refugee camps in Jordan.

### Method

A cross-sectional study was conducted on a convenient sample of forty women diagnosed with GDM from the high-risk maternity clinic in one of the Syrian refugee camps in Jordan. The study used the Gestational Diabetes Management Self-Efficacy Scale (GDMSES), Diabetes Knowledge (DMK) assessment, and Diabetes Self-Care Activities Questionnaire (SDSCA) to measure the variables of interest. Descriptive analysis and Multiple logistic regression were used to assess for significant factors.

### Results

Significant associations were found between the subcategories of diet, exercise, and blood sugar control in both the self-efficacy and self-activity scales ($p < 0.01$, $p < 0.01$, $p < 0.05$), respectively. Two factors were associated with higher GDM self-care: diabetes knowledge and higher self-efficacy toward GDM self-care ($p < 0.05$).

### Conclusion

The findings of this study highlight that pregnant women with GDM who have higher levels of self-efficacy and diabetes knowledge are more likely to achieve higher levels of GDM self-care. Beside developing health promotion programs to enhance women's self-efficacy in adhering to GDM care, adequate support and relevant resources to facilitate GDM management among refugee women are recommended. Future research for identifying other potential factors affecting GDM self-care among refugees is highly recommended.

**Funding:** The authors received no specific funding for this work.

**Competing interests:** The authors have declared that no competing interests exist.

## Introduction

Gestational Diabetes Mellitus (GDM) is glucose intolerance that usually begins or is diagnosed first during pregnancy, and in most cases, it ceases post-delivery [1]. In the United States, it affects yearly around 2–10% of women [2, 3]. In Europe, it was reported to affect 5.4% of women [4]. Furthermore, a high prevalence of GDM (~13%) was found in Middle Eastern countries and North African countries [5]. In Jordan, one study conducted at the Jordan University Hospital from January 2015 to January 2016 showed a high prevalence of (13.5%) of GDM [6].

GDM is found to be associated with an increased risk of adverse fetal outcomes such as neonatal hypoglycemia and macrosomia, as well as maternal complications, including; gestational hypertension, premature rupture of membranes, vaginal candidiasis, and preterm birth [7, 8]. Additionally, women diagnosed with GDM have an increased likelihood of developing type 2 diabetes within 5–10 years post-delivery [9]. Women diagnosed with GDM typically experience a progression through several stages in dealing with the condition, including shock upon receiving the diagnosis, coming to terms with GDM, acquiring new strategies to manage GDM, future planning, and seeking environmental support [10].

Managing GDM can be particularly challenging as it may rely on women's adherence to care [11]. Therefore, it is essential to consider the cultural factors that affect self-care behaviors among women with GDM [12]. The management of GDM includes physical activity; nutritional monitoring and control; glucose monitoring; and pharmacological therapy [13]. Poor self-care behaviors of GDM might increase the risk for comorbidities for both the infant and the mother [10]. Self-care behaviors among women with GDM can be influenced by; multiple socio-demographic, physiological, and psychosocial factors. These factors could facilitate the management, such as; mothers' concerns about their newborn's health and social support, whereas some factors could be barriers to management, for example, physical and social constraints and limited comprehension of GDM management requirements [14, 15]. Based on the social cognitive theory (SCT) [16]. This theory predicts the health behaviour by adopting personal cognitive factors such as self-efficacy (person's belief of the ability to perform behaviour), outcomes expectations (person's belief of the outcomes that result from performing behaviour), and socio-structural factors (facilitators or impediments) [17]. In addition, an adapted model by Shortridge-Baggett [18] suggests person characteristics (e.g. age, gender, knowledge) to be an influencers of health behaviour in people with chronic disease and therefore, this adopted model will guide our research.

Jordan is the sixth largest refugee host country in the world [19]. To escape military conflict in Syria since 2011, millions of Syrians fled to other countries, including Jordan. Jordan hosts over 673 thousand Syrian refugees, 19.5% of whom reside in the United Nations Refugee Agency's three camps; the Za'atari refugee camp is the largest one [20].

Syrian refugees residing in camps in Jordan have lower perceived health compared to other refugees living outside [21]. Healthcare services in these camps are provided by multiple charitable organizations and are offered free of charge [22]. However, some refugees lack access to healthcare services due to structural barriers such as lacking appropriate civil documents [23, 24]. These services focus on communicable and non-communicable diseases, mental health, and women's health, including reproductive health services and the associated issues with child marriages and sexual violence [25]. However, women's health needs and services among Syrian refugees require further attention and improvement [26]. To the best of our knowledge, no studies published to date on GDM among Syrian refugees. Self-management among Syrian refugees was found among patients with type 2 diabetes in Bekaa Valley in Lebanon to find out that knowledge and education related to diabetes, secondary schooling education level, higher

patient self-confidence in self-care as well as insulin use were highly associated with higher scores of diabetes self-management, whereas, a lower score of self-management found to be associated with increasing age and with those who diagnosed during the Syrian conflict [27].

In Turkey, it has been observed that Syrian refugees have lower rates of gestational diabetes testing and antenatal visits. Additionally, Syrian refugee women experience more complicated pregnancies with higher rates of preterm births, rupture of membranes, and low birth weight [28]. Studies conducted in Jordan and Turkey have also shown that Syrian refugee women have significantly higher rates of preterm birth and low birth weight compared to Jordanian and Turkish women, respectively [29, 30]. Furthermore, Boulle and colleagues [31] have identified significant challenges in providing diabetic care by humanitarian organizations in crisis situations, including limited access to essential medicines and diagnostic resources necessary for clinical care. It is crucial to enhance the level of care provided in these circumstances. Limited research is available on the topic of GDM among refugees including Syrian refugees globally [32]; therefore, more research is needed on this topic to provide culturally appropriate diabetes care for these women and improving their pregnancy outcome.

In Jordan, several studies investigated the self-care/ self-management of people with diabetes [33–37]. Al-Khawaldeh et al. [34] found that participants with higher self-efficacy showed better self-management behaviors in diet, blood sugar testing, exercise, and medication. However, there is a scarcity of studies concerning Syrian women and pregnant women with GDM residing in the refugee camps which would help policymakers and donner organizations in prioritizing the health care delivery plans in improving the pregnancy outcomes among this displaced group. Therefore, this study aims to measure the levels of self-care activities of Syrian women with GDM residing in the Za'atari refugee camp and its associated factors looking to the overall concerns of refugee women.

## Methods

### Study design, sampling and setting

A cross-sectional study was conducted on a convenience sample of 40 pregnant women residing in the Za'atari refugee camp in Jordan between September and December 2022. The Za'atari camp, established in 2012, is the largest Syrian refugee camp in Jordan, located approximately 10 kilometers east of Mafraq city in the northern region. The camp hosts over 80,000 residents [20]. By comparing Jordanian and Syrian women in terms of maternal health care, the study revealed that, on average, Syrian women received fewer antenatal care visits when compared to Jordanian women (62% and 82%, respectively) [38].

According to sample size calculation guidelines, it is commonly recommended that for regression analysis, there should be a minimum of 10 observations per variable [39]. As we are utilizing three main variables for regression analysis, a minimum sample size of 30 is required [40]. We recruited further 10 participants (total n = 40) to obtain convenience sample considering potential attrition or missing data. The study included all pregnant women diagnosed with gestational diabetes mellitus (GDM) at a gestational age of more than 20 weeks. However, women with other pregnancy-related comorbidities or pre-existing chronic conditions like Type 1 or Type 2 diabetes mellitus were excluded. Additionally, women who were illiterate or had been diagnosed with mental illness were also excluded from the study.

### Ethical consideration

Prior to administering the questionnaire, a consent form was obtained from each participant, ensuring they were adequately informed about the study. The researcher was available to address any questions or concerns during the process. They were also informed that they

could withdraw it at any time they want. Anonymity and confidentiality were maintained by asking the participants not to mention their names or any information related to them such as: phone numbers, identity documents, and file numbers. All the data were kept within a locked cabinet, and only the researcher had access to this cabinet. The study obtained approval from the Scientific Research Ethical Committee of the Faculty of Nursing at Applied Science Private University (IRB no. Faculty 2021-2022-44) and also received ethical approval from the Ministry of Interior Affairs (no. 5/MT/19166/49608). Furthermore, necessary approvals were obtained from the camp security and clinic management service to conduct the study.

## Data collection and study setting

The study was conducted at the reproductive health clinic-Z3, managed by the Jordan Health and Society (JHASI) at the Za'atari camp in Jordan. This clinic provides comprehensive care for all high-risk pregnant women in the camp, including those diagnosed with GDM. On average, the clinic receives approximately 200 high-risk pregnant women monthly (around 10% of them being diagnosed with GDM). Data collection was carried out by a trained female research assistant who was available to address any inquiries or concerns from the women, considering that these women had only completed primary school education, reflecting the educational background within the camp. Women who met the inclusion criteria (as being pregnant women diagnosed with gestational diabetes mellitus (GDM) at a gestational age of more than 20 weeks were approached by the research assistant at the registration desk in the clinic. The research assistant explained the research objectives and ensured the strict confidentiality of personal information. Permission to participate in this study was obtained from all women. They believe this study would significantly improve their health status and the healthcare services provided to them.

## Study tool

A self-administered questionnaire was utilized in the study, comprising seven sections: *Section One*: Sociodemographic Characteristics—This section captured information regarding age, education, occupation, and socio-economic status. *Section Two*: General Health History and Family History of Diabetes—Participants were asked to provide details about their overall health history and any family history of diabetes. *Section Three*: Pregnancy Information—This section was focused on the current pregnancy, including the Last Menstrual Period (LMP), gestational age, gravidity, parity, activities of daily living (ADLs), and timing and location of antenatal visits. *Section Four*: GDM Profile and Pre-pregnancy BMI—Participants were requested to provide their GDM profile and self-reported BMI before pregnancy, including height in centimeters and weight in kilograms. BMI was calculated using the World Health Organization (WHO) formula and recommended specifications (underweight < 18.5, normal weight 18.5–24.9, pre obesity 25–29.9, obesity >30) [41]. Additionally, it included fasting blood sugar, glucose tolerance test results, gestational age at GDM diagnosis, HbA1C levels which is categorized into two categories (poor self-care >6.5, and good self-care ≤6.5) [42], and blood pressure. *Section Five*: Diabetes Knowledge Questionnaire (DKQ) The Diabetes Knowledge Questionnaire (DKQ) was developed by the researchers based on a literature review assessing different aspects of diabetes knowledge in Arabic [43]—The DKQ consisted of 24 items with multiple-choice questions. Participants received one point for each correct answer and zero for incorrect responses. The total score ranged from (0–24), with higher scores indicating a higher level of diabetes knowledge. The DKQ assessed various aspects, including general knowledge about diabetes (nine items), physical activity (two items), medications (four items), nutrition (three items), complications (two items), foot care (two items),

sick day management (one item), and follow-up and blood sugar monitoring (one item). The DKQ demonstrated good content validity with a Content Validity Index (CVI) of 0.92 and a Cronbach's alpha coefficient of 0.78 this was studied on Jordanian population. *Section Six*: Health Education on GDM—Participants reported whether they received health education on GDM, including topics such as definition, diet management, exercise management, pharmacological care, self-glucose monitoring, and sources of health education. *Section Seven*: Gestational Diabetes Management Self-Efficacy Scale (GDMSES) [44]. This scale, translated into Arabic, consisted of 16 Likert scale items rated from 0 to 10. It measured women's confidence in performing various activities related to GDM management. The scale's score interpretation was as follows: 0–3: low confidence, 4–8: moderate confidence, 9–10: high confidence. The scale exhibited good internal consistency reliability (Cronbach's alpha = 0.85) and stability over time (Pearson correlation coefficient > .6). Section Eight: Diabetes Self-Care Activities (SDSCA) [45]—This instrument assessed the diabetes self-care activities of individuals with diabetes. This instrument assessed the diabetes self-care activities of individuals with diabetes. This study used the Arabic version of (SDSCA), this version was translated into Arabic language using forward and backward translation methods and validated among a diverse range of Arabic-speaking populations.) [46]. The scale evaluated the frequency of performing different self-care activities over the preceding week. Response options ranged from 0 to 7, with higher scores indicating better adherence to self-care activities. The scale comprised 14 items covering seven constructs of diabetes self-care activities: physical activity (2 items), general and specific diet (4 items), blood sugar measurement (2 items), medication (3 items), foot care (2 items), and adherence and smoking behaviors (1 item). A pilot testing on five pregnant women was conducted prior to data collection to make sure that those women can fill the questionnaire without any difficulties of understanding and to estimate the time required to fill the overall questionnaire. However, the women encountered no problem in filling and the time required was between 15–20 minutes, therefore, those women were included in the study.

## Statistical analysis

**Explanatory variables.** The study examined various variables as potential explanatory factors, including sociodemographic characteristics, health-related variables (health history and obstetrical health, GDM-related health variables), and psychosocial variables. Sociodemographic variables: Participants self-reported information on age, marital status, family income, employment, education level, living place, and smoking status. Health-related variables: GDM-related health variables included the type and duration of diabetes, self-reported body mass index (BMI) before pregnancy, HbA1c, glucose tolerance test, gestational age at the time of diagnosis, and details of diabetes treatment. Psychosocial factors examined in the study included participants' knowledge about GDM, self-efficacy in managing GDM, and self-care practices.

**Statistical analysis.** The data were analyzed using IBM SPSS Statistics version 23.0 for Windows (IBM Corp, Armonk, NY, USA). This included descriptive analysis such as frequencies (proportions), mean (standard deviation (SD)), and median (interquartile range (IQR)). Pearson correlation analysis was used to assess the associations between the subclasses of self-efficacy and the subclasses of self-care. Multiple logistic regression analysis was used to examine the association of the subset of sociodemographic variables, health variables, self-efficacy score, and knowledge score (predictors) with the self-care score (outcome). Statistical significance was considered when the p-value was less than 0.05. Regression assumptions were checked using graphical methods (normality of residuals, homoscedasticity), collinearity and cook's distance. Residuals were distributed normally. No signs of multicollinearity were detected.

## Results

A total of 40 pregnant women participated in the study survey. "Table 1" presents the sociodemographic and health characteristics of the participants. The mean age of the women is 33.2 years. The majority of them are unemployed and have completed primary school education. Additionally, they were non-smokers. About 60% of the participants had previously experienced GDM, with an average of 6.7 pregnancies. The majority of women (37 women) had their last checked HbA1C was higher than 6.5mmol/L which indicated poor GDM management. The mean reported pre-pregnancy BMI is 25.2. Regarding their knowledge, the average number of correct answers to the knowledge questions was 10. In addition, more than half of the women (57%) reported receiving health education from the clinic, specifically about GDM. On the one hand, the highest self-efficacy sub-score was related to exercise (mean: 3.91 ± 2.96), indicating a higher level of confidence in this area. On the other hand, the lowest mean score was observed in the diet category (mean: 3.16 ± 2.64) (Table 1).

**Table 1. Characteristics of the study participants (number (%) or mean ±SD) unless otherwise stated\*.**

| Characteristics | Total (n = 40) |
|---|---|
| **Sociodemographic characteristics** | |
| Age (years) | 33.2 ±6.5 |
| Primary school education | 36 (90%) |
| Unemployed | 38 (95%) |
| Number of children* | 5 (3–6) |
| **Health characteristics and activities** | |
| Hypertension | 8 (20%) |
| Thyroid disease | 4 (10%) |
| Previous history of GDM | 24 (60%) |
| Gestational age | 38 (34–39)* |
| Gravida | 6.7 ± 2.7 |
| Para | 4.9 ±2.4 |
| **BMI (before pregnancy)**<br> Underweight: <18.5<br> Normal weight:: 18.5–24.9<br> Pe-obesity: 25–29.9<br> Obesity >30 | **25.2± 3.5**<br>0 (0.0%)<br>19 (47.5%)<br>18 (45%)<br>3 (7.5%) |
| **HbA1C (%, mmol/L)**<br> Poor self-care (>6.5mmol/L)<br> Good self-care (<6.5mmol/L) | **7.3 ±1.6**<br>37(92.5%)<br>2(5%) |
| Regular antenatal following | 36 (90%) |
| **Knowledge on diabetes** | |
| Prior heath education | 23 (57.5%) |
| Knowledge about diabetes | 10 ±5.6 |
| **Self-efficacy (GDMSES)** | |
| Score of Self-efficacy to GD      self-care (GDMSES)<br> Self-efficacy subclasses<br> Diet<br> Exercise<br> Blood control | **1.39 ±1.09**<br>3.16±2.64<br>3.91±2.96<br>3.54 ±2.48 |
| **Score of (SDSCA)** | **1.39 ±1.09** |

\* Median (min-max)

## Diabetes Knowledge Questionnaire (DKQ)

The mean score for the correctly answered questions in the KDM questionnaire was 10 ± 5.6. The highest percentage of correct answers was observed in answering the question related to the best method of measuring blood sugar levels (75%), followed by the question regarding the effect of exercise on blood sugar levels, with a correct response rate of 62%. Conversely, the lowest percentage of correct answers was provided for the question related to the effects of consuming sugary juices on diabetes mellitus (12.5%) "Table 2".

## Gestational Diabetes Self-Care (SDSCA)

"Table 3" presents the mean scores for each item in the GDSS (Gestational Diabetes Self-Management Scale). The highest-scoring activities reported by women in the last 7 days were consuming a high-fat diet and eating fruits and vegetables. Whereas, the activities with the lowest scores were blood sugar testing and engaging in specific exercise sessions.

## Self-efficacy and self-care subclasses correlations

"Table 4" presents the correlation coefficients between the subclasses of the self-efficacy and the subcategories of self-care were examined, revealing significant associations. A strong positive correlation was found between self-efficacy in diet and adherence to dietary self-care (r =

**Table 2. Percentages and frequences of total correct answers in the knowledge questionnaire (KDM).**

| Knowledge scale questions<br>*(Multiple choice 24 questions)* | *N(40)*<br>correct<br>n (%) |
|---|---|
| What is HBA1C | 20 (50.0) |
| What is the normal level of sugar in blood | 18 (45.0) |
| Best method to measure level of sugar | 30 (75.0) |
| What is DM | 11 (27.5) |
| Which of the following is a correct statement regarding DM | 13 (32.5) |
| Choose the signs of high blood sugar | 18 (45.0) |
| Choose signs of low blood sugar | 7 (17.5) |
| Choose reasons for low blood sugar | 10 (25.0) |
| Choose the actions that you should do in case of low blood sugar | 20 (50.0) |
| DM tablet medication effect is | 18 (45.0) |
| Best sites to take insulin injection are | 17 (42.5) |
| Where you should keep insulin vial of pen after use | 19 (47.5) |
| Choose the best statement describe DM medication | 16 (40.0) |
| What you should do when you feel can't eat sufficient food | 14 (35.0) |
| What is the effect of exercise on the level of blood sugar | 25 (62.5) |
| DM patients are advised to perform exercises | 15 (37.5) |
| What is the effect of drinking juices high in sugar | 5 (12.5) |
| Which of the following items never use for treating low blood sugar | 22 (55.0) |
| Choose the best statement about diet restrictions for DM | 15 (37.5) |
| DM patients are advised to check their blood sugar regularly because | 18 (45.0) |
| Which of these problems are not related to DM | 21 (52.5) |
| Choose the complications of DM | 16 (40.0) |
| The reasons why foot complications are noticed among DM patients | 21 (52.5) |
| Best way to make foot care | 11 (27.5) |
| **Total Knowledge score (mean ±SD)** | **10 ±5.4** |

**Table 3. Score of self-care activities of the study participants in the last 7 days (n = 40) (Mean±SD).**

| Items | Mean (SD) |
|---|---|
| **General diet**<br>Item 1: Follow healthȥ diet. | **2.43 (1.83)**<br>1.7 (2.56) |
| Item 2: Follow own diet. | 1.87 (2.56) |
| Item 3: Eats high fat diet. | 3.9 (3.07) |
| Item 4: Eats fruits and vegetables | 2.25 (2.40) |
| **Exercise**<br>Item 5: Participates in at least 30 minutes of physical activity per day | **1.5 (1.65)**<br>2.4(2.75) |
| Item 6: Participates in specific exercise sessions | 0.6 (1.5) |
| **Blood sugar testing**<br>Item 7: Tests for blood sugar | **0.57 (1.23)**<br>0.65 (1.33) |
| Item 8: Tests for blood sugar as recommended by healthcare provider | 0.5 (1.2) |
| **Foot care**<br>Item 9: Checks feet | **0.92 (1.96)**<br>1.1 (2.24) |
| Item 10: Inspects the inside of shoes | 0.75(1.93) |
| **Adherence to treatment**<br>Item 11: Takes diabetes medication | **1.2 (1.55)**<br>1.43(2.62) |
| Item 12: Takes insulin | 0.95 (2.33) |
| Item 13: Takes diabetes pills | 1.45 (2.56) |
| **Smoking**<br>Item 14: Smokes | <br>0.3 (0.16) |

.64, p < 0.01). Similarly, a strong positive correlation was found between self-efficacy in exercise and engagement in exercise-related self-care (r = .67, p < 0.01). Furthermore, a significant positive correlation was identified between self-efficacy in blood control and adherence to blood control self-care (r = .35, p < 0.05).

## Self-care predictors

"Table 5" shows the results of multiple linear regression analysis to test if sociodemographic and health variables, self-efficacy score, and knowledge score significantly predicted self-care scores. The overall regression model was significant ($R^2$ = 0.56, p = 0.006). It was found that the self-efficacy score to SDSCA significantly predicted self-care score (β = 0.458, p = 0.016). It was also found that knowledge about diabetes significantly predicted self-care scores (β = 0.336, p = 0.031). However, sociodemographic and health variables did not significantly predict the self-care score.

**Table 4. Correlation coefficients between self-efficacy subclasses and corresponding self-care subclasses.**

| | | Self-efficacy subclasses | | |
|---|---|---|---|---|
| | | Diet | Blood control | exercise |
| Self-care subclasses | Diet | .637** | .71** | .560** |
| | Blood sugar control | .295ns | .353* | .279ns |
| | exercise | .641** | .62** | .669** |

ns: not significant

*: p < 0.05

**: p < 0.01

**Table 5. Multiple linear regression of sociodemographic selected variables, health variables, self-efficacy score, and knowledge score on self-care.**

| Independent variables | Standardized coefficients beta | significance |
|---|---|---|
| Age | .103 | .556ns |
| Education | -.173 | .606ns |
| Occupation | .156 | .623ns |
| History of GD | -.235 | .159ns |
| Gravida | -.044 | .901 ns |
| Para | -.085 | .928 ns |
| No. of children | -.065 | .944 ns |
| HbA1C | -.066 | .650 ns |
| BMI before pregnancy | .190 | .190 ns |
| Self-efficacy to GDSC | .458 | .016* |
| Knowledge about diabetes | .336 | .031* |
| **Model summary** | | |
| $R^2$ | = .559 | |
| F-ratio | = 3.23 | |
| p-value | P < .006 | |

ns: not significant

*: p < 0.5

## Discussion

The main objective of this study was to assess the levels of self-care and their associations among refugee women with GDM in Za'atari camp, Jordan. The mean score for self-care among the women was 1.39 ± 1.09, while the mean score for self-efficacy was also 1.39 ± 1.09. Two factors were identified to be associated with GDM self-care, which are: self-efficacy and knowledge. Social Cognitive Theory [16] suggests predicting the health behavior of individuals based on two factors: self-efficacy (an individual's belief of their own ability to perform the behavior) and outcomes expectations (an individual's belief of the outcomes that result from performing a health behavior) [17].

The self-care score assesses the activities performed by pregnant women in the past 7 days, with higher scores indicating better adherence to self-care practices. Among the self-care sub-classes, the highest scores were observed in the diet category, followed by adherence to treatment, while the lowest score was found in blood control. These sub- scores were lower compared to the baseline data from an intervention study conducted in Oman, where the highest score was reported for diet (4 ± 1.3), and the lowest score was observed for physical activity (2.5 ± 2.0) [47].

Among the various subscales related to diet adherence, the lowest level of adherence was found in following the prescribed diet plan compared to other sub items. This can be attributed to the lack of health education and support, and limited access to dietary resources especially for refugee women. These women are typically required to follow the same diet as their families, which is primarily supported by the UNHCR food distribution system. A quasi-experimental study conducted among diabetic patients who were Syrian refugees in Jordan demonstrated that a combination of cash assistance and health education was essential for improving effective care and diabetic control. The provision of cash assistance would enable better diet control and facilitate more frequent visits and follow-ups [48].

Blood sugar control (monitoring) was the least performed subscale in terms of self-care, the main reason for that result is that in the clinic, women receive no health education in that regard since GDM women do not receive strips/GlucoCheck kit to be used for self-monitoring for blood sugar level. Therefore, women do not check their blood sugar levels regularly at home, the only chance for them to check their blood sugar is during their antenatal visits, which might take place for high-risk pregnancy women every two to three weeks; however, only patients with DM type 1 or 2 receive strips from the clinic for follow-up and control [48].

Interestingly, the highest score on the knowledge scale was among the question concerning the best method to measure blood sugar (75%) which supports the idea that low self-care activity was related to a lack of resources and affordability. One study among Syrian refugees in Turkey found that only 15% of the participants conducted GDM screening tests since that was costly, not all women afford it, and not covered by humanitarian organizations [28]. Our study found two predictors for GD self-care: self-efficacy and knowledge. Self-efficacy sub-scales for diet, exercise, and blood sugar control/monitoring were highly significant, with similar corresponding subscales. This was similar to a study conducted among DM patients with type 2 self-management for glycemic control in Jordan [49].

Concerning GDM, it was found among other studies worldwide a strong positive relationship between self-efficacy and GDM self-control [50–54]. Self-efficacy was also highly correlated with improving self-management among individuals with DM in several studies [55–60]. Hence, enhancing women's self-efficacy regarding diet control, exercise, and blood sugar control holds the potential to improve their self-management. Patients with high self-efficacy are more inclined to establish goals that facilitate behavioural changes and long-term adherence to new behaviours. This, in turn, may contribute to better overall health outcomes [61]. Based on the social learning theory and cognitive theory, health promotion is greatly influenced by an individual's attitudes and beliefs regarding their ability to perform specific activities correctly. Therefore, the adherence to and maintenance of new health behaviours is more likely to be influenced by stronger self-efficacy [62]. Challenges that refugee women faced during their displacement and adaptation to the current camp life might increase their adaptability to diseases and increase their self-efficacy when resources are available.

Knowledge was another significant predictor found to be associated with self-care among our studied women, knowledge and educational programs were strongly correlated with self-efficacy in other chronic diseases [63]. Therefore, knowledge might be a core effect affecting women's self-efficacy and improving their self-confidence. A study carried out by Saboula and her colleagues [64] in Egypt found that the level of poor knowledge of GDM was 90% among pregnant women, while after an intervention study, it changed to 10%; the study recommended continuous health education for GDM as better knowledge increased self-care activity.

It's worth to mention that refugee's women might encounter other challenges that might influence their self-care, as lack of enough financial resources which might restrict their overall life-style as nutritional care and health care monitoring. Women also encountered problem with education as they all had the basic level of education as restriction of movement as well as financial limitations might hinder their continuing educational dreams that in itself might influence their overall care for their health status. Another challenging factor that faced Syrian women is early marriage, as many of those women had forced to marry at early age where they are not yet ready to tack care of their own health neither their children's care. The overall living condition of women in the refugee camp might create stress and anxiety that might influence the level of GDM and might hinder the self-care activities [65]. More researches are recommended to explore and correlate other factors.

## Strengths and limitations

The novelty of this particular study lies in the fact that Syrian refugee pregnant women with GDM have not been thoroughly researched. The significant factors pertaining to GDM self-care were shown in this study. Although future research could take note of the relatively small sample size, this study's strength in originality persists.

## Conclusion and recommendations

This study showed that women who perceived higher self-efficacy are a potential predictor of better GDM self-care. Refugee women need more attention and support from humanitarian organizations, clinicians, and policymakers to improve women's knowledge and develop new special programs to enhance women's self-efficacy in adherence to GDM care and management, as well as support relevant resources to improve maternal and neonatal outcomes. Furthermore, there is a need for future research to explore refugee women's experience of GDM, their specific needs, and short- and long-term complications of GDM on mothers and their children in the context of humanitarian crisis. This, in turn, can help tailoring individualized and culturally sensitive care for refugees around the globe, which will be reflected on their health.

## Supporting information

**S1 Data.**
(CSV)

## Acknowledgments

We extend our gratitude to the dedicated research assistant who played a crucial role in promptly collecting the data. Special thanks go to all the study participants for their valuable participation and time devoted to completing the questionnaire. Our sincere appreciation and acknowledgment go to the reproductive health clinic-Z3 and its management body, as well as the Jordan Health and Society (JHASI) at Al Za'atari Camp for their cooperation throughout the study. Furthermore, we would like to acknowledge the facilitation provided by the security personnel at the Al Za'atari camp, which contributed to the smooth progress of the research process.

## Author Contributions

**Conceptualization:** Enas A. Assaf, Aaliyah Momani, Rasmieh Al-Amer, Anas Ababneh.

**Data curation:** Enas A. Assaf, Ghada A. Al-Sa'ad.

**Formal analysis:** Enas A. Assaf, Anas Ababneh.

**Investigation:** Enas A. Assaf.

**Methodology:** Enas A. Assaf, Haleama Al Sabbah, Aaliyah Momani, Rasmieh Al-Amer, Ghada A. Al-Sa'ad.

**Project administration:** Enas A. Assaf.

**Visualization:** Aaliyah Momani.

**Writing – original draft:** Enas A. Assaf, Haleama Al Sabbah.

**Writing – review & editing:** Enas A. Assaf, Haleama Al Sabbah, Aaliyah Momani, Rasmieh Al-Amer, Ghada A. Al-Sa'ad, Anas Ababneh.

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
