## [Decision Letter · Decision Letter 0]

16 Oct 2023

PONE-D-23-20978Factors Influencing Gestational Diabetes Self-Care Among Pregnant Women in a Syrian Refugee Camp in JordanPLOS ONE

Dear Dr. Assaf,

Thank you for submitting your manuscript to PLOS ONE. After careful consideration, we feel that it has merit but does not fully meet PLOS ONE’s publication criteria as it currently stands. Therefore, we invite you to submit a revised version of the manuscript that addresses the points raised during the review process.

We look forward to receiving your revised manuscript.

Kind regards,

Nur Aizati Athirah Daud, Ph.D.

Academic Editor

PLOS ONE

**a)** If there are ethical or legal restrictions on sharing a de-identified data set, please explain them in detail (e.g., data contain potentially sensitive information, data are owned by a third-party organization, etc.) and who has imposed them (e.g., an ethics committee). Please also provide contact information for a data access committee, ethics committee, or other institutional body to which data requests may be sent.

3. Please amend the manuscript submission data (via Edit Submission) to include author Ghada A Al-Sa’ad.

**a)** If there are ethical or legal restrictions on sharing a de-identified data set, please explain them in detail (e.g., data contain potentially sensitive information, data are owned by a third-party organization, etc.) and who has imposed them (e.g., an ethics committee). Please also provide contact information for a data access committee, ethics committee, or other institutional body to which data requests may be sent.

3. Please amend the manuscript submission data (via Edit Submission) to include author Ghada A Al-Sa’ad.

Reviewers' comments:

Reviewer's Responses to Questions

**Comments to the Author**

1. Is the manuscript technically sound, and do the data support the conclusions?

Reviewer #1: Partly

Reviewer #2: Partly

2. Has the statistical analysis been performed appropriately and rigorously? 

Reviewer #1: Yes

Reviewer #2: Yes

3. Have the authors made all data underlying the findings in their manuscript fully available?

Reviewer #1: No

Reviewer #2: Yes

4. Is the manuscript presented in an intelligible fashion and written in standard English?

Reviewer #1: No

Reviewer #2: Yes

5. Review Comments to the Author

Reviewer #1: Congratulation to the author and there are rooms for improvement.

Abstract: suggest to add number of participants, and statistical analysis here e.g. any multiple logistic regression used to do the analysis or not.Recommendations for Future Research: Although the conclusion addresses the significance of self-efficacy and diabetes knowledge in GDM self-care, it would be helpful to include recommendations for future research. For instance, identifying other potential factors affecting GDM self-care could be suggested for further investigation.

Introduction

GDM Prevalence: The introduction mentions the prevalence of GDM in the United States, Europe, and Middle Eastern countries but doesn't specifically mention the prevalence in Syrian refugee camps. Consider adding the prevalence of GDM among Syrian refugee women residing in camps in Jordan, if available, to provide a more context-specific perspective.

Rationale for Study in Refugee Camps: The introduction effectively highlights the unique challenges faced by Syrian refugees in accessing healthcare services in the Za’atari refugee camp. However, it would be helpful to explicitly explain why studying GDM and self-care in this specific population is important, such as the potential implications for policy and healthcare delivery in such crisis situations.

The Role of Social Cognitive Theory (SCT): While the introduction briefly mentions SCT and its relevance to predicting health behaviors, consider expanding this section to explain how SCT will be applied in the study and how it relates to the study's focus on self-care behaviors among pregnant women with GDM.

Previous Research: The introduction provides relevant information on previous studies related to self-management and diabetes care among Syrian refugees in other contexts. To strengthen the justification for the current study, consider including more studies or findings related to GDM specifically, or studies conducted in similar refugee camp settings.

Methodology:

Sample Size Justification: suggest to do it or give the reference on minimum of 10 observations per ?? significant variable. The study mentioned that a minimum sample size of 30 is required for regression analysis with three main variables. However, it might be helpful to clarify how the sample size of 40 was determined, considering the convenience sampling method. Explain any considerations for potential attrition or missing data.

Data Collection: The section on data collection is clear and well-organized. However, it might be useful to provide more details on how the questionnaire was administered, especially for illiterate participants. Describe any adaptations or assistance provided to ensure accurate completion of the questionnaire.

Study Tools: The study utilized various validated questionnaires, and their reliability and validity are briefly mentioned. Consider providing additional references or details on the validation studies of the questionnaires, especially those translated into Arabic, to demonstrate their appropriateness for this population.

Ethical Consideration: The section on ethical consideration is well-addressed, with approvals from relevant authorities mentioned. However, it might be helpful to include specific information on how confidentiality and privacy were maintained during data collection and analysis.

Statistical Analysis: The statistical analysis plan is adequately described. However, consider mentioning the assumptions made for multiple linear regression analysis, such as the presence of normality and homoscedasticity. Also, specify how potential confounding factors will be controlled for in the regression model.

Result

Table 1: Clarify the representation of the BMI data. The table shows "Pre-obesity: 25-29.9," but the accompanying text mentions that almost all women were within the normal BMI range. Ensure the correct representation of the BMI categories in the table.

Table 2: In the knowledge questionnaire (KDM) table, consider providing the correct percentage of correct answers for each question instead of the number of participants who answered correctly. What is HBA1C 20 (50.0 ), no need percentage as n(%) already mentioned at the top. Furthermore, this would provide a clearer picture of the participants' overall knowledge.

Table 5: In the regression analysis table, consider providing the beta coefficient and 95%CI on top of p-value for the intercept in the model.

Discussion

Interpretation of Findings: While the discussion provides a good overview of the results, it would be beneficial to provide more detailed explanations of the reasons behind certain findings. For example, discuss in-depth why self-efficacy in blood ??glucose control was low and how this might be related to the lack of access to testing strips and frequent monitoring.

Comparison with Previous Studies: The discussion would be strengthened by comparing the current findings with previous studies in a similar context or with similar populations for levels of self-care result.

The discussion should discuss the potential implications of the findings within the context of refugee camps and the challenges faced by the women in accessing healthcare resources and education. Addressing the socio-economic and cultural factors that impact self-care in this specific setting would enhance the discussion.

Considering the limitations and challenges faced by the refugee women, it would be helpful to propose potential intervention strategies that could improve self-care in this population. Discuss how targeted health education, access to resources, and support systems might positively impact self-efficacy and knowledge.

It would be beneficial to include a section on future research directions that can build on the current study's findings. Suggest potential areas of investigation to improve self-care and health outcomes among refugee women with GDM.

Address the generalizability of the findings to other refugee settings and populations, as well as the potential applicability of the results in developing intervention programs for similar vulnerable groups.

While the limitations of convenience sampling are briefly mentioned, consider elaborating on other potential limitations of the study. Consider mentioning the non-probabilistic nature of convenience sampling and how it may impact the generalizability of the findings. Suggest to discuss on another another limitation on the possibility of response bias or recall bias in self-reported data.

Reviewer #2: 1. Title and Abstract: Your article's title is acceptable, and the abstract provides a concise overview of the content. However, I recommend making sure that the abstract effectively summarizes the key findings and contributions of the article to entice potential readers.

2. Introduction: The introduction effectively sets the stage for the rest of the article by providing context and the rationale for your research. It would be beneficial to include a clearer statement of your research objectives or hypotheses in this section.

3. Literature Review: Your review of the existing literature is comprehensive. It provides a solid foundation for the reader to understand the current state of research on the topic.

4. Methodology: I appreciate the detailed explanation of your research methods, but I suggest providing more clarity regarding the sample size including exclusion & inclusion, data collection process, and data analysis techniques. This will help readers assess the validity and reliability of your findings.

5. Results: Your presentation of the results is clear and easy to follow. To enhance the impact of your article, consider incorporating visual aids such as tables, figures, or graphs (standard to all tables) to support the textual description of your findings.

6. Discussion: The discussion section offers valuable insights and interpretations of the results. However, it would be beneficial to explore the practical implications of your findings in a broader context and suggest areas for future research.

7. Conclusion: No clear conclusion. You might want to emphasize the practical significance of your findings in this section.

8. References: Ensure that your references are consistent in style and formatting throughout the article.

9. Language and Clarity: Thorough proofreading is recommended to eliminate any grammatical errors, and improve sentence structure and overall readability, although the article is well-written.

6. PLOS authors have the option to publish the peer review history of their article (what does this mean?). If published, this will include your full peer review and any attached files.

Reviewer #1: **Yes: **Ching Siew Mooi

Reviewer #2: No

---

## [Author Response · Author response to Decision Letter 0]

27 Oct 2023

Factors Influencing Gestational Diabetes Self-Care Among Pregnant Women in a Syrian Refugee Camp in Jordan

PONE-D-23-20978

Reviewer one:

1. Abstract: suggest to add number of participants, and statistical analysis here e.g. any multiple logistic regression used to do the analysis or not. Recommendations for Future Research: Although the conclusion addresses the significance of self-efficacy and diabetes knowledge in GDM self-care, it would be helpful to include recommendations for future research. For instance, identifying other potential factors affecting GDM self-care could be suggested for further investigation.

Thank you for the comments, number of participants was mentioned (highlighted)

Statistical analysis: multiple logistic regression was added (highlighted).

Conclusion was modified and added this sentence: “Future research for identifying other potential factors affecting GDM self-care among refugees is highly recommended”

2. Introduction

a. GDM Prevalence: The introduction mentions the prevalence of GDM in the United States, Europe, and Middle Eastern countries but doesn't specifically mention the prevalence in Syrian refugee camps. Consider adding the prevalence of GDM among Syrian refugee women residing in camps in Jordan, if available, to provide a more context-specific perspective.

Unfortunately, research on Syrian refugees with GDM is very limited in Jordan and no studies were found on its prevalence

b. Rationale for Study in Refugee Camps: The introduction effectively highlights the unique challenges faced by Syrian refugees in accessing healthcare services in the Za’atari refugee camp. However, it would be helpful to explicitly explain why studying GDM and self-care in this specific population is important, such as the potential implications for policy and healthcare delivery in such crisis situation

This was clarified in the last paragraph of the introduction.

c. The role of Social Cognitive Theory (SCT): While the introduction briefly mentions SCT and its relevance to predicting health behaviors, consider expanding this section to explain how SCT will be applied in the study and how it relates to the study's focus on self-care behaviors among pregnant women with GDM.

this section was further explained as “This theory predicts the health behaviour by adopting personal cognitive factors such as self-efficacy (person’s belief of the ability to perform behaviour), outcomes expectations (person’s belief of the outcomes that result from performing behaviour), and socio-structural factors (facilitators or impediments) (2). In addition, an adapted model by Shortridge-Baggett (3) suggests person characteristics (e.g. age, gender, knowledge) to be an influencers of health behaviour in people with chronic disease and therefore, this adopted model will guide our research”. Highlighted in the text

. D. Previous Research: The introduction provides relevant information on previous studies related to self-management and diabetes care among Syrian refugees in other contexts. To strengthen the justification for the current study, consider including more studies or findings related to GDM specifically, or studies conducted in similar refugee camp settings.

This is presented in the second last paragraph of the introduction. Very limited research on GDM among Syrian refugees is available. There are studies focusing on pregnancy outcomes and possible complications; but they are not specific to women with GDM. Therefore, we conducted the current study. More references were added 

Kragelund Nielsen, K., Davidsen, E., Husted Henriksen, A. and Andersen, G.S., 2023. Gestational diabetes and international migration. Journal of the Endocrine Society, 7(1), p.bvac160. (ref. no. 31)

3. Methodology:

a. Sample Size Justification: suggest to do it or give the reference on minimum of 10 observations per ?? significant variable. The study mentioned that a minimum sample size of 30 is required for regression analysis with three main variables. However, it might be helpful to clarify how the sample size of 40 was determined, considering the convenience sampling method. Explain any considerations for potential attrition or missing data.

We thank the reviewer for this comment.We have added a reference to justify the need of 10 subjects for each variable.

 i.e, Harris RJ. A primer of multivariate statistics: Psychology Press; 2001 (ref. no. 38)

We agree, and we have further explained that: We recruited a further 10 participants (total n=40) to obtain convenience sample considering potential attrition or missing data.

 This was added in the text as “We recruited further 10 participants (total n=40) to obtain convenience sample considering potential attrition or missing data”

b. Data Collection: The section on data collection is clear and well-organized. However, it might be useful to provide more details on how the questionnaire was administered, especially for illiterate participants. Describe any adaptations or assistance provided to ensure accurate completion of the questionnaire.

Thank you for your comments, to explain this point none of the participants were illiterate (this was one of the excluding criteria in the research) they all can read and write, however the researcher assistant were available in the clinic in case there were any questions or clarifications.

c. Study Tools: The study utilized various validated questionnaires, and their reliability and validity are briefly mentioned. Consider providing additional references or details on the validation studies of the questionnaires, especially those translated into Arabic, to demonstrate their appropriateness for this population.

The Diabetes Knowledge Questionnaire (DKQ) was developed by the researchers based on a literature review assessing different aspects of diabetes knowledge . Content validity of DKQ was checked by a panel of four experts who were experts in the fields of health education and diabetes. Content validity index (CVI) was 0.92, and a Cronbach’s alpha of 0.78 was achieved in a study among Diabetes Type 2 in Jordan. 

this was explained in the text (highlighted)

concerning the SCDM scale this was further added to the text “This study used the Arabic version of (SDSCA), this version was translated into Arabic language using forward and backward translation methods and validated among a diverse range of Arabic-speaking populations, keep in mind that this scale demonstrated acceptable reliability of (α= 0.68) [46]”. 

(highlighted)

d. Ethical Consideration: The section on ethical consideration is well-addressed, with approvals from relevant authorities mentioned. However, it might be helpful to include specific information on how confidentiality and privacy were maintained during data collection and analysis.

We clarified this point by adding this to the ethical consideration section “They were also informed that they could withdraw it at any time they want. Anonymity and confidentiality were maintained by asking the participants not to mention their names or any information related to them such as: phone numbers, identity documents, and file numbers. All the data were kept within a locked cabinet, and only the researcher had access to this cabinet” 

e. Statistical Analysis: The statistical analysis plan is adequately described. However, consider mentioning the assumptions made for multiple linear regression analysis, such as the presence of normality and homoscedasticity. Also, specify how potential confounding factors will be controlled for in the regression model.

e. “Regression assumptions were checked using graphical methods (normality of residuals, homoscedasticity), collinearity and cook’s distance. Residuals were distributed normally. No signs of multicollinearity were detected”. (checking model assumptions were added to the text)

We checked the possible confounding effect of age, education, and occupation on self-efficacy and knowledge to GDSC, and found no evidence of possible confounding effect as the coefficients (B) of self-efficacy and knowledge did not change when the possible confounding factors were added to the model. 

4. Result

a. Table 1: Clarify the representation of the BMI data. The table shows "Pre-obesity: 25-29.9," but the accompanying text mentions that almost all women were within the normal BMI range. Ensure the correct representation of the BMI categories in the table.

a.Thank you for the valuable comments, we agree that it was miss explained and the sentence of almost all women were within normal weight was removed.

b. Table 2: In the knowledge questionnaire (KDM) table, consider providing the correct percentage of correct answers for each question instead of the number of participants who answered correctly. What is HBA1C 20 (50.0 ), no need percentage as n(%) already mentioned at the top. Furthermore, this would provide a clearer picture of the participants' overall knowledge.

 b. well considered, and changed thank you 

c. Table 5: In the regression analysis table, consider providing the beta coefficient and 95%CI on top of p-value for the intercept in the model

Please note that the standardized regression coefficients (beta) were already provided in -table 5 under the column (standardized coefficients). We added the word “beta” to make it clearer. Also notice that Spss only provides confidence interval for the unstandardized coefficients, hence CI was not provided.

5. Discussion

a. Interpretation of Findings: While the discussion provides a good overview of the results, it would be beneficial to provide more detailed explanations of the reasons behind certain findings. For example, discuss in-depth why self-efficacy in blood ??glucose control was low and how this might be related to the lack of access to testing strips and frequent monitoring.

a. Thank you for the note, we realized that the interpretation for control was not clear, as control meant to be monitoring for blood sugar, therefore the reasons were clearly stated. This was added to the sentence.

b. Comparison with Previous Studies: The discussion would be strengthened by comparing the current findings with previous studies in a similar context or with similar populations for levels of self-care result.

b.As what was mentioned in the introduction there were scarcity in studies among the Syrian refugees GDM women especially in Jordan, we tried to find similar studies related to DM type 1 and 2, and that’s what we used in the discussion.

c. The discussion should discuss the potential implications of the findings within the context of refugee camps and the challenges faced by the women in accessing healthcare resources and education. Addressing the socio-economic and cultural factors that impact self-care in this specific setting would enhance the discussion.

c. Thank you for the note, we realized that the interpretation for control was not clear, as control meant to be monitoring for blood sugar, therefore the reasons were clearly stated. This was added to the sentence.

d. As what was mentioned in the introduction there were scarcity in studies among the Syrian refugees GDM women especially in Jordan, we tried to find similar studies related to DM type 1 and 2, and that’s what we used in the discussion.

d. Thank you for the valuable comment, this paragraph was added with reference. 

“Its worth to mention that refugee’s women might encounter other challenges that might influence their self-care, as lack of enough financial resources which might restrict their overall life style as nutritional care and health care monitoring. Women also encountered problem with education as they all had the basic level of education as restriction of movement as well as financial limitations might hinder their continuing educational dreams, that in itself might influence their overall care for their health status. Another challenging factor that faced Syrian women is early marriage, as many of those women had forced to marry at early age where they are not yet ready to tack care of their own health neither their children’s health. The overall living condition of women in the refugee camp might create stress and anxiety that might influence the level of GDM and might hinder the self-care activities [65]. More researches are recommended to explore and correlate other factors”

6. Considering the limitations and challenges faced by the refugee women, it would be helpful to propose potential intervention strategies that could improve self-care in this population. Discuss how targeted health education, access to resources, and support systems might positively impact self-efficacy and knowledge.

It would be beneficial to include a section on future research directions that can build on the current study's findings. Suggest potential areas of investigation to improve self-care and health outcomes among refugee women with GDM.

Address the generalizability of the findings to other refugee settings and populations, as well as the potential applicability of the results in developing intervention programs for similar vulnerable groups.

While the limitations of convenience sampling are briefly mentioned, consider elaborating on other potential limitations of the study. Consider mentioning the non-probabilistic nature of convenience sampling and how it may impact the generalizability of the findings. Suggest to discuss on another another limitation on the possibility of response bias or recall bias in self-reported data.

Thanks for your comment. Based on our current study we cannot propose a specific intervention to improve GDM management. However, we can direct future research to focus on this matter. This is considered in the conclusion section. As this sentence was added “their specific needs, and short- and long-term complications of GDM on mothers and their children in the context of humanitarian crisis. This, in turn, can help tailoring individualized and culturally sensitive care for refugees around the globe which will be reflected on their health.” And this would serve generalizability as well.

Reviewer two

Title and Abstract: Your article's title is acceptable, and the abstract provides a concise overview of the content. However, I recommend making sure that the abstract effectively summarizes the key findings and contributions of the article to entice potential readers.

Thank you for your comment, abstract was modified especially the conclusion section

Introduction: The introduction effectively sets the stage for the rest of the article by providing context and the rationale for your research. It would be beneficial to include a clearer statement of your research objectives or hypotheses in this section.

Thank you appreciate your comment

Methodology: I appreciate the detailed explanation of your research methods, but I suggest providing more clarity regarding the sample size including exclusion & inclusion, data collection process, and data analysis techniques. This will help readers assess the validity and reliability of your findings.

We appreciate your comments, all you mentioned is now clearer and included.

Results: Your presentation of the results is clear and easy to follow. To enhance the impact of your article, consider incorporating visual aids such as tables, figures, or graphs (standard to all tables) to support the textual description of your findings.

Thank you , included

Discussion: The discussion section offers valuable insights and interpretations of the results. However, it would be beneficial to explore the practical implications of your findings in a broader context and suggest areas for future research.

Thank you , this was added as well by the end of the discussion paragraph

Conclusion: No clear conclusion. You might want to emphasize the practical significance of your findings in this section.

Thank you , this was updated and changed

References: Ensure that your references are consistent in style and formatting throughout the article.

Thank you , checked

Language and Clarity: Thorough proofreading is recommended to eliminate any grammatical errors, and improve sentence

Thank you , this was rechecked and re- edited .

---

## [Decision Letter · Decision Letter 1]

14 Dec 2023

PONE-D-23-20978R1Factors Influencing Gestational Diabetes Self-Care Among Pregnant Women in a Syrian Refugee Camp in JordanPLOS ONE

Dear Dr. Assaf,

Thank you for submitting your manuscript to PLOS ONE. After careful consideration, we feel that it has merit but does not fully meet PLOS ONE’s publication criteria as it currently stands. Therefore, we invite you to submit a revised version of the manuscript that addresses the points raised during the review process.

We look forward to receiving your revised manuscript.

Kind regards,

Nur Aizati Athirah Daud, Ph.D.

Academic Editor

PLOS ONE

Journal Requirements:

Reviewers' comments:

Reviewer's Responses to Questions

**Comments to the Author**

1. If the authors have adequately addressed your comments raised in a previous round of review and you feel that this manuscript is now acceptable for publication, you may indicate that here to bypass the “Comments to the Author” section, enter your conflict of interest statement in the “Confidential to Editor” section, and submit your "Accept" recommendation.

Reviewer #2: All comments have been addressed

Reviewer #3: All comments have been addressed

2. Is the manuscript technically sound, and do the data support the conclusions?

Reviewer #2: Yes

Reviewer #3: Yes

3. Has the statistical analysis been performed appropriately and rigorously? 

Reviewer #2: Yes

Reviewer #3: Yes

4. Have the authors made all data underlying the findings in their manuscript fully available?

Reviewer #2: Yes

Reviewer #3: Yes

5. Is the manuscript presented in an intelligible fashion and written in standard English?

Reviewer #2: Yes

Reviewer #3: Yes

6. Review Comments to the Author

Reviewer #2: Line 38: Multiple logistic regression

Line 216: ... keep in mind that this scale) - hanging sentence

Line 242: Multiple linear regression

*please correct whether linear or logistic

Table 1: characteristics - Knowledgee on diabetes? or general health (please correct)

Line 279 - typo Gestational Diabetes ..... (please correct)

Line 303: R superscript 2

I propose considering professional proofreading services or engaging in peer reviews to thoroughly review the manuscript.

Reviewer #3: Thank you for addressing all the comments well. The article looks much better now with clear objectives and learning points.

7. PLOS authors have the option to publish the peer review history of their article (what does this mean?). If published, this will include your full peer review and any attached files.

Reviewer #2: No

Reviewer #3: **Yes: **NAVIN KUMAR DEVARAJ

---

## [Author Response · Author response to Decision Letter 1]

22 Dec 2023

Reviewer comments Response

Reviewer #2

Line 38: Multiple logistic regression 

Well considered, thank you .

Line 216: ... keep in mind that this scale) - hanging sentence 

Thank you for the review. This sentence was deleted.

Line 242: Multiple linear regression

*please correct whether linear or logistic 

Thank you , this was changed to multiple logistic regression

Table 1: characteristics - Knowledgee on diabetes? or general health (please correct)

 Thank you , this was Corrected to Knowledge on diabetes 

Line 279 - typo Gestational Diabetes ..... (please correct)

 Corrected , thank you 

Line 303: R superscript 2

 changed

I propose considering professional proofreading services or engaging in peer reviews to thoroughly review the manuscript. 

Well considered, Proof reading was done.

Reviewer #3

Thank you for addressing all the comments well. The article looks much better now with clear objectives and learning points.

 Thank you for your valuable time to review our work, we highly appreciate that.

---

## [Editor Report · Decision Letter 2]

28 Dec 2023

Factors Influencing Gestational Diabetes Self-Care Among Pregnant Women in a Syrian Refugee Camp in Jordan

PONE-D-23-20978R2

Dear Dr. Assaf,

We’re pleased to inform you that your manuscript has been judged scientifically suitable for publication and will be formally accepted for publication once it meets all outstanding technical requirements.

Kind regards,

Nur Aizati Athirah Daud, Ph.D.

Academic Editor

PLOS ONE
---

## [Editor Report · Acceptance letter]

2 Feb 2024

PONE-D-23-20978R2 

PLOS ONE

Dear Dr. Assaf, 

I'm pleased to inform you that your manuscript has been deemed suitable for publication in PLOS ONE. Congratulations! Your manuscript is now being handed over to our production team.

Kind regards, 

on behalf of

Dr. Nur Aizati Athirah Daud 

Academic Editor

PLOS ONE